# Willingness to Receive the COVID-19 and Seasonal Influenza Vaccines among the Saudi Population and Vaccine Uptake during the Initial Stage of the National Vaccination Campaign: A Cross-Sectional Survey

**DOI:** 10.3390/vaccines9070765

**Published:** 2021-07-08

**Authors:** Amel Ahmed Fayed, Abeer Salem Al Shahrani, Leenah Tawfiq Almanea, Nardeen Ibrahim Alsweed, Layla Mohammed Almarzoug, Reham Ibrahim Almuwallad, Waad Fahad Almugren

**Affiliations:** Department of Clinical Sciences, College of Medicine, Princess Nourah bint Abdulrahman University, Riyadh 13412, Saudi Arabia; aafayed@pnu.edu.sa (A.A.F.); Almanea.l.t77@gmail.com (L.T.A.); Harnard222@gmail.com (N.I.A.); Almarzouq.layla@gmail.com (L.M.A.); Rihamsaeed4444@gmail.com (R.I.A.); Waadfahad1@hotmail.com (W.F.A.)

**Keywords:** COVID-19, seasonal influenza, vaccine, uptake, pandemic, Saudi Arabia

## Abstract

This study aimed to assess the willingness to receive the coronavirus disease 2019 (COVID-19) and seasonal influenza vaccines and vaccine uptake during the early stage of the national vaccination campaign in Saudi Arabia. A cross-sectional online survey was conducted among adult Saudis between 20 January and 20 March 2021. The questionnaire addressed vaccine hesitancy, perceived risk, willingness, and vaccine uptake. Approximately 39% of the participants expressed vaccine hesitancy, and 29.8% and 24% felt highly vulnerable to contracting COVID-19 and seasonal influenza, respectively. The majority (59.5%) were willing to receive the COVID-19 vaccine, although only 31.7% were willing to receive the flu vaccine. Adjusted analysis showed that vaccine hesitancy (OR 0.34, 95% CI 0.27–0.43) and the perception of being at high risk (OR 2.78, 95% CI 1.68–4.60) independently affected the intention to be vaccinated. Vaccine hesitancy was similar among those who were willing to be vaccinated (29.8%) and those who had already been vaccinated (33.1%). The perceived risk was significantly higher among those who had been vaccinated (48.1%) than among those who were willing to be vaccinated but had not yet been vaccinated (29.1%). In conclusion, the acceptance of the COVID-19 vaccine in Saudi Arabia is high. Saudis who received the vaccine had a similar level of vaccine hesitancy and a higher level of perceived risk.

## 1. Introduction

Seasonal influenza spreads globally each year, causing significant morbidity and mortality among different age groups. In 2018, influenza caused three to five million cases of severe illness and 290,000–650,000 deaths worldwide [1]. The coronavirus disease 2019 (COVID-19) pandemic has spanned the season in which influenza is prevalent. The virus that causes COVID-19, severe acute respiratory syndrome coronavirus 2 (SARS-CoV-2) has transmission characteristics similar to those of influenza viruses, including airborne droplets and direct contact with infected individuals [2]. In March 2020, the World Health Organization (WHO) declared that the COVID-19 outbreak was a pandemic. Since then, SARS-CoV-2 has caused at least 124,894,108 infections and 2,745,702 deaths [3].

Moreover, the COVID-19 pandemic has imposed a major burden on healthcare facilities and significantly disrupted societies and economies worldwide. However, border closures, travel restrictions, and other precautionary measures led to dramatic reductions in the detection of cases of influenza in 2020 [4]. In the Eastern Mediterranean region, out of 6998 patients tested for influenza, only 8.3% tested positive [5]. Despite these figures, there is growing concern about SARS-CoV-2 circulating concurrently with seasonal influenza viruses in the absence of sufficient data on the risk of coinfection with the two viruses [6,7].

Vaccination is the most effective way to prevent infectious diseases and has helped to eradicate diseases such as poliomyelitis [8]; it also reduced the severity of illness linked to the influenza A pandemic [9]. Thus, the current situation regarding COVID-19 has prompted scientific experts worldwide to accelerate the development of vaccines to help control the pandemic, especially given the lack of curative treatments and herd immunity [10]. The first mass vaccination program started in early December 2020, and as of 15 February 2021, 175.3 million vaccine doses have been administered globally [11]. However, willingness to be vaccinated against COVID-19 is not merely dependent on vaccine efficacy and safety; other contributing factors, such as personal perceptions and hesitancy about vaccination in general and, more specifically, about vaccination against COVID-19, should be addressed before launching mass vaccination campaigns. One systematic review identified that the highest COVID-19 vaccine acceptance rates were reported in Ecuador (97.0%), Malaysia (94.3%), Indonesia (93.3%), and China (91.3%). Moderate to low rates of vaccine acceptance were reported in France (58.9%), the USA (56.9%), Russia (54.9%), Italy (53.7%), Jordan (28.4%), and Kuwait (23.6%) [12].

Saudi Arabia (SA) has a population of 34,218,169, of which 70% is distributed over three main regions (central, eastern, and western Saudi Arabia). Moreover, 72% of the population is between 15 and 64 years [13]. As of 27 March 2021, there were 387,292 reported cases of COVID-19, with a 2.20% mortality rate, and approximately 4 million doses of COVID-19 vaccines had been administered [14]. A recent national study on 1000 individuals showed that the acceptance rate was 64.7% [15], and another study showed that 48% of Saudi individuals accepted the vaccine [16]. The intention to receive the COVID-19 vaccine can be extrapolated from stated willingness and the uptake of the flu vaccine; however, willingness to be vaccinated does not guarantee future vaccination. Thus, in this study, we aimed to assess the willingness to be vaccinated against COVID-19 and seasonal influenza and compared the results with the uptake of both vaccines during the early stage of the national vaccination campaign in Saudi Arabia.

## 2. Materials and Methods

### 2.1. Study Design, Participants, and Setting

We conducted an anonymous cross-sectional online survey among adult Saudis from the 20 January to 20 March 2021. During the study period, the Ministry of Health (MOH) in Saudi Arabia began implementing a vaccination program against COVID-19, starting with healthcare professionals and vulnerable populations, and then expanding it to all residents in all regions. Thus, by the time data collection was complete, immunization was available to all residents of Saudi Arabia. The actual vaccine coverage in Saudi Arabia from Our World in Data is shown in Figure 1 [17].

The study data were collected and managed using the Research Electronic Data Capture (REDCap) electronic data capture tools hosted at Princess Nourah bint Abdulrahman University (PNU). REDCap is a secure, web-based software platform designed to support data capture for research studies [18]. A standardized questionnaire was designed based on a literature review. There were no restrictions on participation based on gender, nationality, or location within Saudi Arabia, and all adults who could access the questionnaire via smartphones or personal computers could complete the survey.

### 2.2. Sample Size and Sampling Technique

This was a cross-sectional study using a snowball convenience sampling technique. Initially, the study investigators shared the survey link with their primary contacts (aged 18 and above) via social networks (Facebook Menlo Park, CA, USA); Twitter (San Francisco, CA, USA) and WhatsApp (Mountain View, CA, USA) and the official e-mail service at PNU. The initial participants were asked to forward the survey to their connections, and so on. To ensure the distribution of the survey to different regions in Saudi Arabia, we recruited data collection facilitators in three regions that had low response rates. Because of the self-selected and non-probabilistic nature of the sample, invitations and response rates were not quantifiable according to the American Association for Public Opinion Research reporting guidelines [19].

The estimated sample size needed was 980, based on a power of 0.95, and the expected prevalence of the intention to be vaccinated against COVID-19 and seasonal influenza was 35%, with ±5% as the margin of error. Data collection continued until we reached the estimated sample size plus 40% to compensate for any incomplete surveys.

### 2.3. Study Outcomes and Variables

The questionnaire addressed: (1) demographic characteristics (age, gender, area of residence, education, occupation, average monthly income, smoking and chronic medical conditions), (2) vaccine hesitancy, (3) perceived risk of contracting seasonal influenza/COVID-19, and (4) willingness to be vaccinated against seasonal influenza and COVID-19.

We evaluated participants’ self-reported vaccine hesitancy according to the WHO definition using three questions: “Have you ever refused a vaccine for yourself or a child because you considered it to be useless or dangerous?” “Have you ever postponed a vaccine recommended by a physician because of doubts about it?” “Have you ever accepted a vaccine for a child or yourself despite doubts about its efficacy”. If a participant answered yes to one of these questions, he or she was considered “vaccine hesitant” [20].

The item “What is the likelihood that you will contract COVID-19/seasonal influenza?” was answered on a five-point Likert scale (1 = very unlikely, 2 = unlikely, 3 = neutral, 4 = likely, and 5 = very likely) and was used to examine the participants’ perceived risk.

The item “To what extent you would like to receive a vaccination if it is available?” was answered on a five-point Likert scale (1 = very unlikely, 2 = unlikely, 3 = neutral, 4 = likely, and 5 = very likely) and was used to examine the participants’ willingness to be vaccinated. We then converted the item into a dichotomous scale, where original scores of 1 to 3 were recoded as low willingness to be vaccinated and original scores of 4 or 5 were recoded as high willingness.

Participants were asked to rate the trustworthiness of different sources of information about the vaccines using a five-point scale (5 = very trustworthy, 4 = trustworthy, 3 = neutral, 2 untrustworthy, and 1 = very untrustworthy). These sources of information included general websites, official medical websites such as the Saudi MOH website, social media, family/friends, and healthcare professionals. For analytical purposes, scores of 4 or 5 were aggregated into one group of trustworthy resources, and scores of 3, 2, and 1 were aggregated into another category of untrustworthy resources.

We investigated vaccination uptake by directly asking participants if they had received the vaccine or not and recorded their response as yes or no. The participants were not asked to present any medical certifications or official confirmation of immunization status.

### 2.4. Statistical Analysis

We used SPSS version 21 (IBM Corp. Released 2012. IBM SPSS Statistics for Windows, Version 21.0. Armonk, NY: IBM Corp.) to analyze the data. Descriptive analysis was used to describe the demographic characteristics. The chi-square test was used to compare the perceived risk and vaccine hesitancy between individuals who were willing to be vaccinated against COVID-19 and those who had al-ready been vaccinated. We used logistic regression analysis to identify the factors significantly associated with the willingness to be vaccinated (against COVID-19 and seasonal influenza). The associations are reported as the crude odds ratios (CORs) and adjusted odds ratios (AORs) with 95% confidence intervals (CIs) after adjustment for confounders, including sociodemographic and occupational factors. All tests were two-tailed, and *p*-values less than 0.05 were considered statistically significant.

## 3. Results

A total of 1539 participants completed the survey, with 901 (58.6%) females and 136 (9.1%) individuals 50 years old or older. Most participants were from the central region (*n* = 672; 43.8%) and had a university degree (*n* = 999; 65%), and only 10.9% worked in the health sector. The majority were nonsmokers, and 15.2% reported having chronic diseases. Table 1 lists all the characteristics of the study population.

Approximately 39% of the participants were vaccine hesitant, as defined by the WHO criteria. Regarding the perceived risk, 29.8% and 24% felt highly vulnerable to contracting COVID-19 and seasonal influenza, respectively. The majority of the respondents (59.5%) were willing to receive the COVID-19 vaccine, but only 31.7% were willing to receive the seasonal influenza vaccine when it became available. Healthcare professionals were ranked as the most trustworthy source of information about the vaccine (73.9% of respondents), followed by medical websites such as the Saudi MOH (60.7%) (Table 2).

The willingness to be vaccinated against COVID-19 differed across groups stratified by demographic characteristics; older participants and males were more willing to be vaccinated than younger adults and females, while education and economic levels did not affect willingness to be vaccinated. However, the effects of the demographic characteristics of the participants were attenuated in the adjusted analysis, showing that vaccine hesitancy (AOR 0.34, 95% CI 0.27–0.43), a high perceived risk (AOR 2.78, 95% CI 1.68–4.60), and former smoking (AOR 1.57, 95% CI 1.05–2.35) independently explained the willingness of the participants to be vaccinated against COVID-19 (Table 3).

The willingness to be vaccinated against seasonal influenza was significantly associated with the participants’ education and economic levels; participants who had completed only minimal schooling and those who had received a university education were less willing to be vaccinated than the participants who had completed postgraduate studies. Moreover, willingness to be vaccinated was associated with a higher perceived risk of contracting influenza. However, the adjusted regression analysis showed that a high perceived risk (AOR = 3.55, 95% CI 2.05–6.13), vaccine hesitancy (AOR = 0.45, 95% CI 0.36–0.57), and former smoking (AOR = 2.35, 95% CI 1.40–3.95) were the only factors significantly associated with the willingness to be vaccinated against seasonal influenza (Table 3).

Out of 1539 participants, 127 (8.3%) and 300 (19.5%) had already been vaccinated against COVID-19 and seasonal influenza, respectively. We compared vaccine hesitancy and perceived risk among those who were not willing to be vaccinated against COVID-19, those who were willing but had not yet been vaccinated and those who had already been vaccinated. Vaccine hesitancy was significantly greater among those who were not willing to be vaccinated (56.2%) than among those who were willing to be vaccinated (29.8%) and those who had already been vaccinated (33.1%). The level of perceived risk was significantly higher among those who had already been vaccinated (48.1%) than among those who were willing but had not yet been vaccinated (29.1%) and those who were not willing to be vaccinated (26.6%). The level of vaccine hesitancy was comparable between those who were willing but had not yet been vaccinated and those who had already been vaccinated (Figure 2).

## 4. Discussion

This study aimed to investigate the level of COVID-19 vaccine acceptance and uptake among Saudis just after safe and effective vaccines became available. We found that the majority of Saudis were likely to opt to be vaccinated against COVID-19. Vaccine hesitancy and complacency negatively affect vaccine uptake; however, a relatively similar level of vaccine hesitancy was noted between those who were willing to be vaccinated and those who had already been vaccinated, and interestingly, a higher level of perceived risk was observed among those who had already been vaccinated.

In the current study, two-thirds of the participants were willing to be vaccinated against COVID-19, but only one-third intended to be vaccinated against seasonal influenza. The COVID-19 pandemic has had a massive global impact and has become a constant focus of media attention; the continuous media coverage of the number of cases and deaths and the lack of effective pharmaceutical interventions combined with excitement about the effectiveness of vaccines and hope that it may become possible to return to “normal” life can explain the higher proportion of participants who intend to be vaccinated against COVID-19 than against the seasonal influenza.

An uneven level of acceptance of the COVID-19 vaccine among Saudis was reported before the vaccines were available and remained throughout the early period after the arrival of effective vaccines in the country [15,16]. This is understandable, as the intention to be vaccinated is affected by time, the severity of the pandemic, media coverage, and the extent of restrictions. The intention to be vaccinated tends to be higher during the lockdown stages than during the reopening stages, as people have an increased sense of the real physical and psychological threats posed by COVID-19 [21].

The intention to be vaccinated against COVID-19 and seasonal influenza increased as the perceived risk of contracting COVID-19 or seasonal influenza increased. This finding is in line with those of many other studies from Saudi Arabia and other countries. Remarkably, we found that people who already been vaccinated had significantly higher levels of the perceived risk of contracting COVID-19 than did those who were not willing to be vaccinated and even than those who were intending to be vaccinated. These results confirm the role of perceived risk in judgments and decision making regarding a disease such as COVID-19, which is associated with serious outcomes, an unpredictable disease course, and the limited availability of therapeutic interventions. In contrast, irrespective of the data about the severity of illness following infection, the perceived risk associated with seasonal influenza is likely to be relatively lower, as people are aware of the illness and have even experienced it; it is a common illness, and it is not catastrophic [22], particularly when compared to COVID-19.

It is believed that smokers are hesitant to be vaccinated in general, and a small number of studies have evaluated their intentions regarding vaccination against COVID-19. In the current study, former smokers were more likely to accept vaccination against COVID-19 and influenza than nonsmokers. Current smokers did not show significant reluctance to be vaccinated. These findings partially agree with a recent study from the United Kingdom that showed that smokers had a more negative attitude towards the COVID-19 vaccine than nonsmokers, while former smokers were more willing to receive the vaccine against COVID-19 than current smokers or nonsmokers [23].

Vaccine hesitancy was reported among 38.6% of the participants in the current study; moreover, among those who had already been vaccinated, 33.1% were considered vaccine hesitant. Any interpretation of this finding should take into consideration the fact that vaccine hesitancy is context specific and varies across individuals, places, times, and vaccines [24]. The WHO-SAGE defined vaccine hesitancy as a “delay in [the] acceptance or refusal of vaccination despite [the] availability of vaccination services”; vaccine hesitancy is considered one of the greatest global public health threats in the 20th century and has been a major roadblock to the tremendous efforts that have been made to control infectious diseases [25]. During the current COVID-19 pandemic, vaccine hesitancy is understandable, given the novelty of the disease, concerns about vaccine safety and efficacy due to its unusually rapid development [26], and, likely, the circulation of misleading information [27] and conspiracy theories [28] about the pandemic. Interestingly, the participants who had already been vaccinated and those who intended to be vaccinated had comparable levels of vaccine hesitancy; however, only approximately half of those who were not willing to receive the vaccine were also considered vaccine hesitant. This result highlights the importance of vaccine hesitancy as a real barrier to widespread vaccination and indicates the urgent need to share with this group information about the safety and effectiveness of the vaccine and the relevant legislation and regulations enacted by governments and scientific communities to encourage them to overcome their hesitancy.

In addition to the vaccine acceptance, vaccine uptake is critical issue. The gap between intention and observed behavior is a well-known dilemma with regard to health behaviors. In the current study, only 8% had already been vaccinated against COVID-19, and 19.5% had been vaccinated against influenza. When we compared the levels of uptake of both vaccines to international statistics, both were low. One review article reported that among the general population, the acceptance rates of the influenza and COVID-19 vaccines were 69% and 77.6%, respectively [29]. Another study conducted among nurses reported uptake rates of 49% and 63% for the influenza and COVID-19 vaccines, respectively [30]. Therefore, it is important to consider our results since the data were collected during the first phase of the COVID-19 vaccination campaign, in which priority was given to healthcare workers, elderly individuals, and patients with obesity, chronic diseases and autoimmune diseases [31]. This phase continued until March 2021, after the end of the seasonal influenza immunization campaign. Furthermore, COVID-19 vaccine uptake is very dynamic, and the trend in uptake in Saudi Arabia mirrors the global increase in uptake.

One of the important factors that may influence vaccine uptake is trust in the source of information, especially as the current pandemic has given rise to a significant misinfodemic [32]. Most of the study sample (73.9%) trusted healthcare professionals’ recommendations, followed by scientific/governmental websites (60.7%). Similar findings were reported among the US population [33]. This finding also emphasizes the important role of healthcare professionals in combatting pandemics, whether through vaccination promotion at the institutional/community level or as role models at the personal level. In our study, only 10% of the sample were healthcare professionals; thus, it is difficult to estimate vaccination willingness and hesitancy in this population. However, a recently published national study showed that 50.5% of healthcare professionals were willing to be vaccinated against COVID-19, while 50.29% were delaying vaccination until further safety information was released [34].

Saudi Arabia occupies a unique position as the location of the most holy sites for Muslims worldwide. As has happened every year for 14 centuries, Muslim pilgrims gather in Makkah to perform important rituals. This pilgrimage is the religious high point of a Muslim’s life and an event in which every Muslim dream of participating. In 2019, the number of Muslims traveling to Saudi Arabia for religious reasons exceeded 19 million [35]. The COVID-19 pandemic posed challenges for these pilgrims; restrictions and precautionary measures were implemented in the country to protect both residents and visitors. The Saudi MOH has expanded the coverage of vaccination against COVID-19 to all populations across the country, including citizens and noncitizens, and efforts are continuing to reach the goal of vaccinating 80% of the population [36]. Efforts are now focused on reaching candidates for vaccination through primary health care centers, secondary and tertiary hospitals, new vaccination centers, and even pharmacies, which are well distributed across all regions.

The creation of legislative actions, common international public health policies and even mandates requiring the vaccination of adults—which have not been widely used [37,38]—might be adopted by local authorities, especially given the opening of international borders and the start of the pilgrimage (Hajj) season [39].

The current study has some strengths, such as the large sample size; representativeness of the study population, which was drawn from various geographical regions and demographic backgrounds; and the unique timing of the study, which coincided with the arrival of COVID-19 vaccines in the country and the initiation of the national immunization campaign.

Nevertheless, we are aware of the limitations of this study, including the cross-sectional design, the nonprobability sampling technique, and the online distribution of the questionnaire, which may have limited the representativeness of the sample population. However, according to data from The World Bank, the percentage of Saudis who use the internet surpassed 95% in 2019. Despite this high percentage, neither the demographic distribution nor the usage pattern is known in detail. This broad coverage has motivated many researchers to collect data online; however, this could introduce selection bias [40]. Another weakness is the small number of participants who had already been vaccinated against COVID-19; nonetheless, it is worth noting that studying this small number of participants addresses the gap between intention and uptake with regard to vaccination, and it seems that better uptake than predicted will occur in Saudi Arabia.

## 5. Conclusions

During the early stage of the national immunization campaign, acceptance of the COVID-19 vaccine increased. Saudis who had already received the vaccine had a similar level of general vaccine hesitancy as those who were willing to be vaccinated, and the former perceived a higher level of risk of contracting COVID-19 than both those who were not willing to be vaccinated and those who were willing but had not yet been vaccinated.

## Figures and Tables

**Figure 1 vaccines-09-00765-f001:**
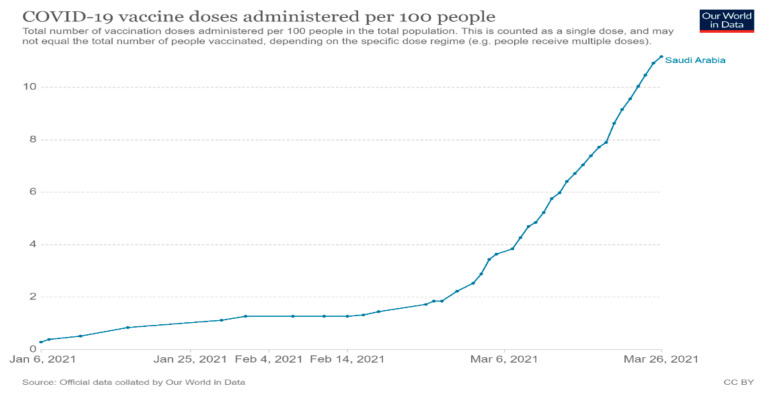
Vaccine coverage during the study period in Saudi Arabia, from Our World in Data.

**Figure 2 vaccines-09-00765-f002:**
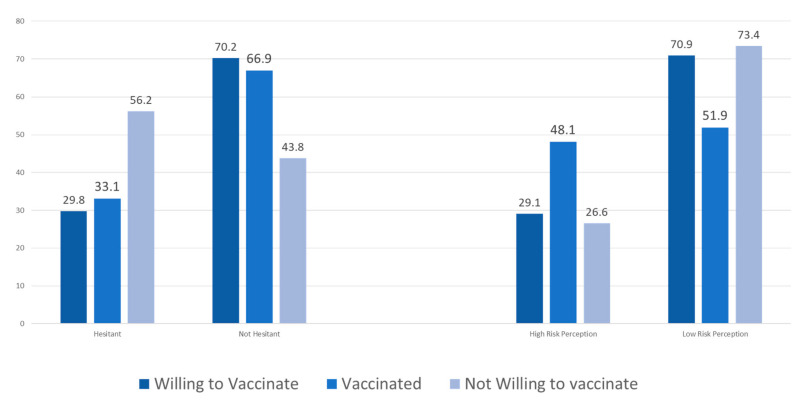
Comparison of perceived risk and vaccine hesitancy among participants who were not willing to be vaccinated against COVID-19, those who were willing but had not yet been vaccinated and those who had already been vaccinated.

**Table 1 vaccines-09-00765-t001:** Characteristics of the study population.

Variable	*n* (%)
**Gender**	
Female	901 (58.6)
Male	636 (41.4)
**Age**	
Under 30 years	788 (52.7)
30–49 years	570 (38.2)
50+ years	136 (9.1)
**Region of residence**	
Central	672 (43.8)
East	236 (15.4)
West	329 (21.4)
North	190 (12.4)
South	109 (7.1)
**Education**	
Minimal schooling (1–12 years)	310 (20.2)
University	999 (65.0)
Higher education (postgraduate)	228 (14.8)
**Occupation**	
Student	588 (38.3)
Employed	638 (41.5)
Retired/unemployed	311 (20.2)
**Average monthly income**	
Below 10,000 SR	443 (29.0)
10,000–15,000 SR	429 (28.1)
15,000–20,000 SR	276 (18.1)
More than 20,000 SR	377 (24.7)
**Smoking**	
Current smoker	208 (13.6)
Former smoker	87 (5.7)
Nonsmoker	1239 (80.8)
**Chronic diseases**	**234 (15.2)**
Diabetes	81 (5.3)
Hypertension	84 (5.5)
Bronchial asthma	72 (4.7)
Cardiac diseases	22 (1.4)

**Table 2 vaccines-09-00765-t002:** Vaccine hesitancy and intention to be vaccinated among the study population.

Vaccine Hesitancy	*n* (%)
Have you ever refused a vaccine for yourself or a child because you considered it to be useless or dangerous?	286 (18.6)
Have you ever postponed a vaccine recommended by a physician because of doubts about it?	293 (19.1)
Have you ever accepted a vaccine for a child or yourself despite doubts about its efficacy?	302 (19.7)
Overall vaccine hesitancy	594 (38.6)
**Perceived risk**	
COVID-19	
Very likely	189 (12.3)
Likely	266 (17.5)
Fair	668 (43.9)
Unlikely	262 (17.2)
Very unlikely	138 (9.1)
Seasonal influenza	
Very likely	119 (7.7)
Likely	250 (16.3)
Fair	688 (44.8)
Unlikely	343 (22.3)
Very unlikely	137 (8.9)
**Willingness to be vaccinated**	
COVID-19	909 (59.5)
Seasonal influenza	486 (31.7)
**Vaccination received**	
COVID-19	127 (8.3)
Seasonal influenza	300 (19.5)
**Trustworthiness of different sources of vaccine-related information**	
General websites	294 (19.1)
Official Saudi Ministry of Health website	888 (60.7)
Family/friends	362 (24.9)
Social media	394 (27.1)
Healthcare professionals	1089 (73.9)

**Table 3 vaccines-09-00765-t003:** Factors associated with willingness to be vaccinated against COVID-19 and seasonal influenza among the study population.

Willingness to Be Vaccinated against COVID-19	COR (95% CI)	AOR (95% CI)
Age	1.01 (1.00–1.02) *	1.02 (0.99–1.03)
Gender (Ref: female)	1.55 (1.24–1.94) *	1.17 (0.89–1.56)
**Education** (Ref: postgraduate)		
School	1.15 (0.79–1.67)	1.07 (0.67–1.70)
University	0.89 (0.65–1.20)	0.89 (0.61–1.31)
**Occupation** (Ref: employed)		
Student	0.84 (0.66–1.07)	1.29 (0.90–1.84)
Unemployed/retired)	0.84 (0.63–1.12)	0.86 (0.61–1.23)
**Economic status (Ref: more than 20,000 Riyals/year)**		
Less than 10,000 riyals/year	0.77 (0.57–1.03)	0.74 (0.52–1.03)
10,000–15,000 riyals/year	1.03 (0.76–1.39)	1.03 (0.73–1.44)
15,000–20,000 riyals/year	0.84 (0.60–1.18)	0.82 (0.57–1.17)
**Smoking (Ref: nonsmoker)**		
Former smoker	1.78 (1.26–2.52) *	1.57 (1.05–2.35) *
Smoker	1.76 (1.06–2.94) *	1.58 (0.90–2.78)
**Chronic diseases (Ref: No)**	1.32 (0.97–1.80)	1.04 (0.73–1.49)
**Vaccine hesitancy (Ref: No)**	0.34 (0.27–0.42) *	0.34 (0.27–0.43) *
**Perceived risk (Ref: very unlikely)**		
Unlikely	1.25 (0.82–1.89)	1.16 (0.75–1.82)
Fair	2.45 (1.68–3.56) *	2.66 (1.78–3.98) *
Likely	2.02 (1.32–3.09) *	2.06 (1.30–3.26) *
Very likely	2.77 (1.73–4.43) *	2.78 (1.68–4.60) *
**Willingness to be vaccinated against seasonal influenza**	COR (95% CI)	AOR (95% CI)
Age	1.01 (0.99–1.01)	1.00 (0.99–1.02)
Gender (Ref: female)	0.84 (0.68–1.05)	1.03 (0.79–1.33)
**Education (Ref: postgraduate)**		
Minimal schooling	0.37 (0.25–0.55) *	1.26 (0.82–1.92)
University	0.55 (0.39–0.79) *	1.04 (0.73–1.49)
**Occupation (Ref: employed)**		
Student	0.89 (0.70–1.14)	0.81 (0.58–1.13)
Unemployed/retired	0.79 (0.59–1.05)	0.86 (0.61–1.19)
**Economic level (Ref: more than 20,000 riyals/year)**		
Less than 10,000 riyals/year	0.64 (0.47–0.86) *	0.87 (0.64–1.20)
10,000–15,000 riyals/year	0.73 (0.54–0.99) *	0.86 (0.63–1.18)
15,000–20,000 riyals/year	1.09 (0.76–1.54)	0.81 (0.57–1.13)
**Smoking (Ref: nonsmoker)**		
Former smoker	1.65 (1.15–2.08) *	2.35 (1.40–3.95) *
Smoker	2.65 (1.64–4.27) *	1.41 (0.99–1.20)
**Chronic diseases (Ref: No)**		0.77 (0.57–1.07)
**Vaccine hesitancy (Ref: No)**	0.45 (0.36–0.57) *	0.45 (0.36–0.57) *
**Perceived risk (Ref: very unlikely)**		
Unlikely	1.34 (0.89–2.02)	1.41 (0.91–2.19)
Fair	1.79 (1.23–2.63) *	1.90 (1.26–2.86) *
Likely	2.95 (1.92–4.56) *	3.11 (1.95–4.94) *
Very likely	3.15(1.89–5.27) *	3.55(2.05–6.13) *

* *p*-value < 0.05.

## Data Availability

All data used in the study are available to interested researchers upon request from the corresponding author after approval from the Institutional Review Board at PNU (contact irb@pnu.edu.sa).

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
