# Peer review of "Willingness to Receive the COVID-19 and Seasonal Influenza Vaccines among the Saudi Population and Vaccine Uptake during the Initial Stage of the National Vaccination Campaign: A Cross-Sectional Survey"

_vaccines, 2021, doi:10.3390/vaccines9070765_

Round 1

Reviewer 1 Report

The introduction read well but I had some questions about the methodology. In Line 7 they write : "starting with healthcare professionals and vulnerable populations, and then expanding to all residents in all regions." More detail is needed. How were these two populations defined and how did recruitment occur? Also, more detail is needed about the snowball sampling technique used and the "sources of information" question. . The authors also write that they used regression analyses to determine predictors of willingness (Line 130) but then later state that they adjusted for confounders (line 132). Was this a prediction model or was a specific association of interest being explored? More is needed in terms of explaining the modeling approach.

The results section also needs some clarifications. The authors write that this was a reported trust worthy source: "browsing medical websites such as the Saudi Ministry of Health" First, the website is the source not the browsing. Second, were respondents asked in detail about the websites they used so to justify that statement? Table 3 is not well summarized. There were several independent factors in the crude analysis that were statistically significant and not mentioned.  The results in lines 168-177 were never mentioned in the statistical part of the methods section. 

This study could benefit by providing the reading with insight on the representativeness of the study sample? Did it generally reflect the county's population at large? How many have access to the internet? This can be articulated under limitations (or strengths).

The Discussion is final over all. 

Minor comments:

  • WHO should be spelled out when initially introduced.
  • Line 79 seems to be missing a parenthesis
  • demographical should be demographic
  • Line 37 should read, "...the virus has caused at least.." since by the time this paper is published the numbers will be higher.
  • In some instances, commas are used with large numbers and in other instances it is not.  Similarly, some %s have one decimal point and others do not. Be consistent.
  • Spell out PNU.
  • Line 144 starts sentence with the as opposed to The
  • Line 146 has health-care as opposed to healthcare

Author Response

Dear Reviewer ,

Point-by-point response to reviewers

Reviewer Number

Original comments of the reviewer

Reply by the author(s)

Changes done on page number and line number

Methods

In Line 7 they write: "starting with healthcare professionals and vulnerable populations, and then expanding to all residents in all regions." More detail is needed. How were these two populations defined and how did recruitment occur?

This sentence referred to the priority list for COVID-19 National vaccination campaign.

The sentence is rephrased to be clearer

Page 2

Lines 76-79

Also, more detail is needed about the snowball sampling technique used and the "sources of information" question.

Done, details about snowball technique were added as well as source of information question

Page 3

Lines 94-100

The authors also write that they used regression analyses to determine predictors of willingness (Line 130) but then later state that they adjusted for confounders (line 132). Was this a prediction model or was a specific association of interest being explored? More is needed in terms of explaining the modeling approach.

Done,

Predictors were replaced by “significant factors that can explain the willingness to get the vaccine. Adjustments of possible confounders as sociodemographic and occupational factors were considered in building the models.

The models were constructed with willing or not to get the vaccines as the outcome (dependent variable) and all factors that showed statistically significant association with willingness to vaccinate in the univariate analysis (unadjusted odds ratio) were included as independent factors in the models to estimate the significant association (adjusted odds ratio) after adjustment of all possible confounders.

Page 4

Lines 140-145

Results

The authors write that this was a reported trustworthy source: "browsing medical websites such as the Saudi Ministry of Health" First, the website is the source not the browsing. Second, were respondents asked in detail about the websites they used so to justify that statement?

Done, “browsing” was deleted.

Yes, the participants were asked to rate each source of information

Page 4

Lines 130-133

Table 3 is not well summarized.

There were several independent factors in the crude analysis that were statistically significant and not mentioned.  The results in lines 168-177 were never mentioned in the statistical part of the methods section

Done, more details about the significant crude adjustments were added.

In the statistical analysis section this paragraph was added “Comparison of participants who were willing to receive the COVID-19 vaccine and those who had already received the vaccine according to their perceived risk of susceptibility and hesitancy towards vaccination was done using Chi-square test”

Page 4

Lines 140-143

This study could benefit by providing the reading with insight on the representativeness of the study sample. Did it generally reflect the county's population at large? How many have access to the internet? This can be articulated under limitations (or strengths).

In Saudi Arabia, the percentage of individuals using the internet exceeded 95% in 2019 according to the data reported from The World Bank. Despite this high percentage, the demographic distribution and usage pattern is not reported in detail. This good coverage encouraged many researchers to use online data collection but still this can carry some sort of selection bias.

https://data.worldbank.org/indicator/IT.NET.USER.ZS?locations=SA

This paragraph has been added to the strengths and limitation section.

Minor comments

- WHO should be spelled out when initially introduced.

- Line 79 seems to be missing a parenthesis

demographical should be demographic

- Line 37 should read, "...the virus has caused at least.." since by the time this paper is published the numbers will be higher.

- In some instances, commas are used with large numbers and in other instances it is not.  Similarly, some %s have one decimal point and others do not. Be consistent.

- Spell out PNU.

- Line 144 starts sentence with the as opposed to The

- Line 146 has healthcare as opposed to healthcare

All minor comments were fixed as requested, changes on the manuscript done accordingly and highlighted in yellow.

Many thanks for your valuable comments. Kindly find attached reply form 

Best Regards,

Reviewer 2 Report

This manuscript presents  a cross section study of vaccine hesitancy for two vaccines in SaudiA rabia      Comments here are meant to improve the quality of the reporting 

It is not clear what new information is pesented here that is not in other pzpers   there is no set of gaps in the literzture znd the studies cited seem to include the same informztion as is presented in the paper    

it is not cleaar as to whether t]he title describes the data    i think that uptake is not measured, and the title and measures section should calrify this nd be accurzte   

anayses and results are fine and well organizied.    Discussion should discuss the hypothees as they will be presented in the introduction 

Author Response

Dear Reviewer ,

Point-by-point response to reviewers

Reviewer Number

Original comments of the reviewer

Reply by the author(s)

Changes done on page number and line number

Comments

It is not clear what new information is pesented here that is not in other pzpers there is no set of gaps in the literzture znd the studies cited seem to include the same informztion as is presented

The gap in the literature was the discrepancy between willingness to vaccinate and the actual uptake of the vaccine that cannot be investigated until the availability of effective vaccine. Additionally, the level of vaccine acceptance is a dynamic process that fluctuates according to the severity of the pandemic, the news about the effectiveness of available vaccines and personal experiences of vaccination. So, our research problem was focused mainly on the level of vaccine acceptance during the initial stage of national vaccination campaign and how people who got vaccinated may differ from those who are willing or not to get vaccinated.

This paragraph is written in the introduction.

“The intention to receive the COVID-19 vaccine can be extrapolated from willingness and the uptake of the flu vaccine; however, willingness to take the vaccine does not guaran-tee vaccination. Thus, in this study, we aimed to assess the willingness to take the COVID-19 and seasonal influenza vaccines and compared the results with the uptake of both vaccines during the early stage of the national vaccination campaign in Saudi Arabia.”

Page 2

Lines 69-74

it is not clear as to whether the title describes the data i think that uptake is not measured, and the title and measures section

should clarify this and be accurate

  • The title is rephrased to:

“Willingness to Receive the COVID-19 and Seasonal Influenza Vaccines among the Saudi Population and their Uptake during the Initial Stage of the National Vaccination Campaign: A Cross-Sectional Survey”

·         The vaccine uptake refers to the actual vaccination of participants in this study during the early stage of immunization campaign. We did not plan to measure the uptake of the vaccine, we planned to compare between people who got the vaccine and those who were willing or not to get it.  The actual uptake of the vaccine according to the official authorities during the study time was cited in the methods section for more credibility.

This paragraph was added to the measures section

“We investigated vaccination uptake by directly asking participants if they received the vaccine or not, and recorded their response as yes or no. Participants were not needed to present any medical certifications or official confirmation of immunization status.”

Page 1

Actual uptake of COVID-19 vaccine is shown in figure 1

Page 4, lines 134-136

analyses and results are fine and well organized. Discussion should discuss the hypotheses as they will be presented in the introduction

The discussion section was focused on the factors that affect the vaccine acceptance, uptake and the actual situation in Saudi Arabia. A special consideration of the timing of the study and the Saudi plans to increase the vaccine uptake was highlighted.

Many thanks for your valuable comments . Please find attached reply form 

Best Regards, 

Round 2

Reviewer 1 Report

The authors have been responsive to all comments.

Reviewer 2 Report

This is improved by the authors and no further comments are needed